# ResIST: Layer-Wise Decomposition of ResNets for Distributed Training

**Chen Dun**[1]     **Cameron R. Wolfe**[1]     **Christopher M. Jermaine**[1]     **Anastasios Kyrillidis**[1]

[1]Computer Science Dept., Rice University, Houston, Texas, USA

## Abstract

We propose `ResIST`, a novel distributed training protocol for Residual Networks (ResNets). `ResIST` randomly decomposes a global ResNet into several shallow sub-ResNets that are trained independently in a distributed manner for several local iterations, before having their updates synchronized and aggregated into the global model. In the next round, new sub-ResNets are randomly generated and the process repeats until convergence. By construction, per iteration, `ResIST` communicates only a small portion of network parameters to each machine and never uses the full model during training. Thus, `ResIST` reduces the per-iteration communication, memory, and time requirements of ResNet training to only a fraction of the requirements of full-model training. In comparison to common protocols, like data-parallel training and data-parallel training with local SGD, `ResIST` yields a decrease in communication and compute requirements, while being competitive with respect to model performance.

## 1 INTRODUCTION

**Background.** The field of Computer Vision (CV) has seen a revolution, beginning with the introduction of AlexNet during the ILSVRC2012 competition. Following this initial application of deep convolutional neural networks (CNNs), the introduction of the residual connection (ResNets) allowed scaling to massive depths without being crippled by issues of unstable gradients during training [He et al., 2016b]. The capabilities of ResNets have been further expanded in recent years, but the basic ResNet architecture has remained widely-used. While ResNets have become a standard building block for the advancement of CV research, the computational requirements for training them are signifi-

cant. For example, training a ResNet50 on ImageNet with a single NVIDIA M40 GPU takes 14 days [You et al., 2018].

Distributed training with multiple GPUs is commonly adopted to speed up the training process for ResNets. Yet, such acceleration is achieved at the cost of a remarkably large number of GPUs (e.g 256 NVIDIA Tesla P100 GPU in [Goyal et al., 2017]). Additionally, frequent synchronization and high communication costs create bottlenecks that hinder such methods from achieving speedups with respect to the number of available GPUs [Shi et al., 2018]. Asynchronous approaches avoid the cost of synchronization, but stale updates complicate their optimization process [Assran et al., 2020]. Other methods, such as data-parallel training with local SGD [Stich, 2019, Lin et al., 2018, Zhang et al., 2016, McMahan et al., 2017], reduce the frequency of synchronization. Similarly, model-parallel training has gained in popularity by decreasing the cost of local training between synchronization rounds [Ben-Nun and Hoefler, 2019, Zhu et al., 2020, Kirby et al., 2020, Gunther et al., 2020, Guan et al., 2019, Chen et al., 2018a].

**This paper.** We focus on efficient distributed training of CNNs with residual skip connections. Our proposed methodology accelerates synchronous, distributed training by leveraging ResNet robustness to layer removal [Huang et al., 2016]. In particular, a group of high-performing subnetworks (sub-ResNets) is created by partitioning the layers of a shared ResNet model to create multiple, shallower sub-ResNets. These sub-ResNets are then trained independently (in parallel) for several iterations before aggregating their updates into the global model and beginning the next iteration. Through the local, independent training of shallow sub-ResNets, this methodology both limits synchronization and communicates fewer parameters per synchronization cycle, thus drastically reducing communication overhead. We name this scheme *ResNet Independent Subnetwork Training* (`ResIST`). The contributions of this work are:

- We propose a distributed training scheme for ResNets, dubbed `ResIST`, that partitions the layers of a global

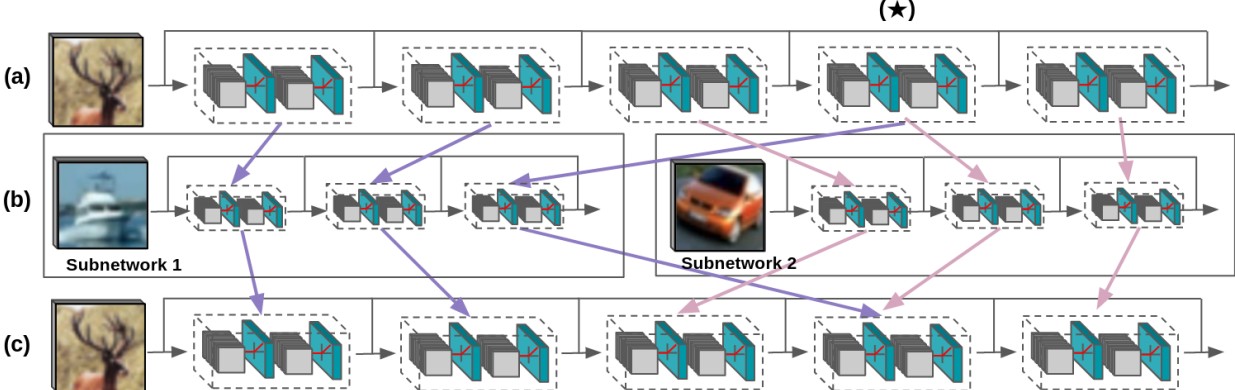

Figure 1: The ResIST model: **Row** (**a**) represents the original global ResNet. **Row** (**b**) shows the creation of two sub-ResNets. Observe that subnetwork 1 contains the residual blocks #1, #2 and #4, while subnetwork 2 contains the residual blocks #3, #4 and #5. **Row** (**c**) shows the reassembly of the global ResNet, after locally training subnetworks 1 and 2 for some number of local SGD iterations; residual blocks that are common across subnetworks (e.g., residual block #4, marked with a ⋆) are aggregated appropriately during the reassembly.

model to multiple, shallow sub-ResNets, which are then trained independently between synchronization rounds.

- We provide theory that ResIST (based on simple ResNet architectures) converges linearly, up to an error neighborhood, using distributed gradient descent with local iterations. We show that the behavior of ResIST is controlled by the overparameterization parameter $m$, as well as the number of workers $S$ in the distributed setting, the number of local iterations, as well as the depth $H$ of the ResNet architecture. Such findings reflect practical observations that are made in the experimental section.

- We perform extensive ablation experiments to motivate the design choices for ResIST, indicating that optimal performance is achieved by $i$) using pre-activation ResNets, $ii$) scaling intermediate activations of the global network at inference time, $iii$) sharing layers between sub-ResNets that are sensitive to pruning, and $iv$) imposing a minimum depth on sub-ResNets during training.

- ResIST is shown to achieve high accuracy and time efficiency in all cases. We conduct experiments on several image classification and object detection datasets, including CIFAR10/100, ImageNet, and PascalVOC.

- We utilize ResIST to train numerous different ResNet architectures (e.g., ResNet101, ResNet152, and ResNet200) and provide implementations for each in PyTorch [Paszke et al., 2019].

## 2 SUB-RESNET TRAINING

ResIST operates by partitioning the layers of a global ResNet to different, shallower sub-ResNets, training those independently, and intermittently aggregating their updates into the global model. The high-level process followed by

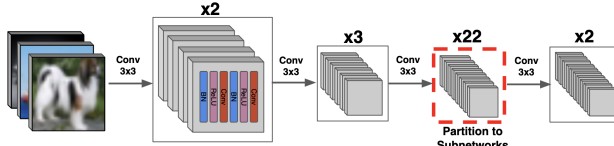

Figure 2: The ResNet101 model used in the majority of experiments. The figure identifies the convolutional blocks that are partitioned to subnetworks. The plot depicts the pre-activation ResNet setting, where we use BN, ReLU, and Conv layers twice in sequence. The network is comprised of four major "sections", each containing a certain number of convolutional blocks of equal channel dimension.

ResIST is depicted in Fig. 1 and outlined in more detail by Algorithm 1. *We note that a naive, uniform partitioning of blocks to each subnetwork, resembling a distributed implementation of [Huang et al., 2016], performs poorly (see Figure 1 in the Appendix ).* To improve upon this procedure, extensive design choices, outlined in section A in the Appendix, are studied to motivate ResIST, leading to a final methodology that generalizes well across domains and datasets.

### 2.1 MODEL ARCHITECTURE

To achieve optimal performance with ResIST, the global model must be sufficiently deep. Otherwise, sub-ResNets may become too shallow after partitioning, leading to poor performance. For most experiments, a ResNet101 architecture is selected, which balances sufficient depth with reasonable computational complexity. Experiments with deeper architectures are provided in section A.4 in the Appendix.

`ResIST` performs best with pre-activation ResNets [He et al., 2016a]. Intuitively, applying batch normalization prior to the convolution ensures that the input distribution of remaining residual blocks will remain fixed, even when certain layers are removed from the architecture. The Pre-activation ResNet101, which we utilize for the majority of experiments, is depicted in Fig. 2. This model, as well as deeper variants (e.g., ResNet152 and ResNet200), are readily available through deep learning packages like PyTorch [Paszke et al., 2019] and Tensorflow [Abadi et al., 2015].

## 2.2 SUB-RESNET CONSTRUCTION

Pruning literature has shown that strided-, initial-, and final-layers within CNNs are sensitive to pruning [Li et al., 2017]. Additionally, repeated blocks of identical convolutions (i.e., equal channel size and spatial resolution) are less sensitive to pruning [Li et al., 2017]. Drawing upon these results, `ResIST` only partitions blocks within the third section of the ResNet (see the highlighted section in Fig. 2), while all other blocks are shared between sub-ResNets. These blocks are chosen for partitioning because $i$) they account for the majority of layers; $ii$) they are not strided; $iii$) they are located within the middle of the network (i.e., initial/final layers are excluded); and $iv$) they reside within a long chain of identical convolutions. By partitioning these blocks, `ResIST` allows sub-ResNets to be shallower than the global model, while maintaining high performance.

The process of constructing sub-ResNets follows a simple procedure; see Figure 1. From row $(a)$ to $(b)$ within Figure 1, indices of partitioned layers within the global model are randomly permuted and distributed to sub-ResNets in a round-robin fashion. Each sub-ResNet receives an equal number of convolutional blocks (e.g., see row $(b)$). In cases, residual blocks may be simultaneously partitioned to multiple sub-ResNets to ensure sufficient depth (e.g., see $(\star)$ in Figure 1). `ResIST` produces subnetworks with $\mathcal{O}(\frac{1}{S})$ of the global model depth, where $S$ is the number of independently-trained sub-ResNets.[1] To contrast this with existing non-distributed attempts, stochastic depth networks [Huang et al., 2016] have an expected depth of 75% of the global model.

The shallow sub-ResNets created by `ResIST` accelerate training and reduce communication in comparison to methods that communicate and train the full model. Table 1 shows the comparison of local SGD to `ResIST` with respect to the amount of data communicated during each synchronization round for different numbers of machines, highlighting the superior communication-efficiency of `ResIST`.

---

[1]A fixed number of blocks is excluded from partitioning (i.e., blocks not in the third section). As a result, this approximation of $\mathcal{O}(\frac{1}{S})$ becomes more accurate as the network becomes deeper (i.e., deeper ResNet variants only add blocks to the third section), as a larger ratio of total blocks are included in the partitioning process.

Table 1: Data communicated during each communication round (in GB) of both local SGD [Stich, 2019] and `ResIST` across different numbers of machines with ResNet101.

| Method | 2 Machine | 4 Machine | 8 Machine |
|---|---|---|---|
| Local SGD | 0.662 GB | 1.325 GB | 2.649 GB |
| ResIST | **0.454** GB | **0.720** GB | **1.289** GB |

## 2.3 DISTRIBUTED TRAINING

The `ResIST` training procedure is outlined in Algorithm 1. Sub-ResNet construction (i.e., `subResNets(·)` in Algorithm 1) follows the procedure outlined in Sec. 2.2. After constructing the sub-ResNets, they are trained independently in a distributed manner for $\ell$ iterations. Following independent training, the updates from each sub-ResNet are aggregated into the global model. Aggregation (i.e., `aggregate(·)` in Algorithm 1) sets each global network parameter to its average value across the sub-ResNets to which it was partitioned. If a parameter is only partitioned to a single sub-ResNet, aggregation simplifies to copying the parameter into the global model. After aggregation, the global model is re-partitioned randomly to create a new group of sub-ResNets, and this entire process is repeated.

---

**Algorithm 1** RESIST Meta Algorithm

---
**Parameters**: $T$ synchronization iterations, $S$ sub-ResNets, $\ell$ local iterations, $\mathcal{W}$ ResNet weights.
$h(\mathcal{W}) \leftarrow$ randomly initialized ResNet.
**for** $t = 0, \ldots, T-1$ **do**
   $\{h_s(W_s)\}_{s=1}^S = $ `subResNets`$(h(W), S)$.
   Distribute each $h_s(W_s)$ to a different worker.
   **for** $s = 1, \ldots, S$ **do**
     // Train $h_s(W_s)$ for $\ell$ iterations using local SGD.
     **for** $l_t = 1, \ldots, \ell$ **do**
       $W_s = W_s - \eta \frac{\partial L}{W_s}$
     **end for**
   **end for**
   $h(\mathcal{W}) = $ `aggregate` $\left(\{h_s(W_s)\}_{s=1}^S\right)$.
**end for**

---

## 2.4 BASELINE CHOICE

Common baselines for distributed training are generally split into data- and model-parallelism protocols. Focusing on the former, the communication efficiency of `ResIST` significantly surpasses data-parallelism. In particular, data parallel methods need to synchronize the whole model at every training iteration, while `ResIST` only needs to communicate the weights of sub-ResNets among the workers.

Typically, model parallel techniques split the model into modules (such as layers) and distribute these modules to

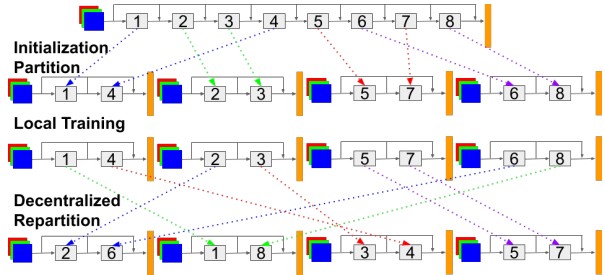

Figure 3: A depiction of the decentralized repartition procedure. This example partitions a ResNet with eight blocks into four different sub-ResNets. The "blue-green-red" squares dictate the data that lies per worker; the orange column dictates the last classification layer. As seen in the figure, each worker is responsible for only a fraction of parameters of the whole network. The whole ResNet is never fully stored, communicated or updated on a single worker.

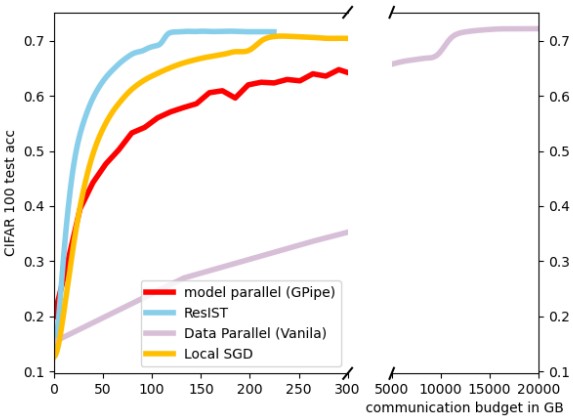

Figure 4: Communication efficiency of ResIST versus data parallelism (vanila), model parallelism (GPipe - [Huang et al., 2019]) and local SGD (LSGD) on CIFAR100.

each worker. At every training iteration, input data is first passed to the worker containing the network's beginning module (e.g., the first layer). Then, at each module, the worker $i$) performs a forward pass of its module and $ii$) sends the resulting output activation to the worker containing the next module. After the last module is activated, the final loss and gradient is calculated before the backward pass is performed, where each worker receives gradient information needed for updating module weights.

Model-parallelism often suffers from higher communication frequency and volume, in comparison to data parallel methods, due to the significant cost of transmitting network activation maps between workers. E.g., model parallel training of ResNets requires transmission of the full batch activation map between layers, which is more cumbersome than simply communicating network parameters. ResIST is more

communication efficient compared to common model parallel methods (e.g., GPipe [Huang et al., 2019]).

Within this work, we adopt local SGD [Lin et al., 2018]— a strong variant of data parallel training—as our baseline. Similar to ResIST, local SGD performs local training iterations on each worker between synchronizations, thus largely decreasing communication frequency and volume. To justify this selection, we perform a baseline comparison, which is displayed in Figure 4 and further detailed in Sections 2.5 and 5. As shown in Figure 4, ResIST is significantly more communication efficient in comparison to data-parallelism (vanilla) and model-parallelism (GPipe), thus making local SGD a more appropriate baseline.

## 2.5   IMPLEMENTATION DETAILS

ResIST is implemented in PyTorch [Paszke et al., 2019], using the NCCL communication package. We use basic broadcast and reduce operations for communicating blocks in the third section and all reduce for blocks in other sections. We adopt the same communication procedure for the local SGD baseline to ensure fair comparison. *The implementation of ResIST is decentralized, meaning that it does not assume a single, central parameter server.*

As shown in Figure 3, during the synchronization and repartition step following local training, each sub-ResNet will directly send each of its locally-updated blocks to the designated new sub-ResNet. Each worker will only need sufficient memory to store a single sub-ResNet, thus limiting the memory requirements. Such a decentralized implementation allows parallel communication between sub-ResNets, which leads to further speedups by preventing any single machine from causing slow-downs due to communication bottlenecks. The implementation is easily scalable to eight or more machines, either on nodes with multiple GPUs or across distributed nodes with dedicated GPUs.

ResIST reduces the number of bits communicated at each synchronization round and accelerates local training with the use of shallow sub-ResNets. The authors are well-aware of many highly-optimized versions of data-parallel and synchronous training methodologies [Paszke et al., 2019, Abadi et al., 2015, Sergeev and Del Balso, 2018]. ResIST is fully compatible with these frameworks and can be further accelerated by leveraging highly-optimized distributed communication protocols at the systems level, which we leave as future work. Further, the authors are well-aware of advanced recent decentralized distributed computing techniques as in [Koloskova et al., 2020, Nedic and Ozdaglar, 2009, Assran and Rabbat, 2020, Koloskova et al., 2019]; our aim is to show the benefits of our approach even on simpler distributed frameworks, and we leave the extension of ResIST to such more advanced protocols as future work.

## 2.6 SUPPLEMENTAL TECHNIQUES

**Scaling Activations.** Similar to [Huang et al., 2016], activations must be scaled appropriately to account for the full depth of the resulting network at test time. To handle this, the output of residual blocks in the third section of the network (see Figure 2) is scaled by $1/S$, where $S$ is the number of sub-ResNets. Such scaling allows the global model to perform well, despite using all layers at test time.

**Subnetwork Depth.** Within `ResIST`, sub-ResNets may become too shallow as the number of sub-ResNets increases. To solve this issue, `ResIST` enforces a minimum depth requirement, which is satisfied by sharing certain blocks between multiple sub-ResNets. Through experimental analysis, a minimum of five blocks partitioned to each sub-ResNet was found to perform optimally. Such a finding motivates our choice of the ResNet101 architecture, as ResNet50 contains only five blocks for partitioning. `ResIST` is extensible to deeper architectures; see section A.4 in the Appendix.

**Tuning Local Iterations.** We use a default value of $\ell = 50$, as $\ell < 50$ did not noticeably improve performance. In some cases, the performance of `ResIST` can be improved by tuning $\ell$ (see Figure 2 in Appendix). The optimal $\ell$ setting in `ResIST` is further explored in section A.3 in the Appendix.

**Local SGD Warm-up Phase.** Directly applying `ResIST` may harm performance on some large-scale datasets (e.g., ImageNet). To resolve this, we perform a few epochs with data parallel local SGD before training the model with `ResIST`.[2] By simply pre-training a model for a few epochs with local SGD, the remainder of training is completed using `ResIST` without a significant performance decrease.

## 3 THEORETICAL RESULT

We provide proof that the gradient descent direction of combined updates from all sub-ResNets, during distributed local training, is close to the hypothetical gradient descent direction of the whole model as if trained centrally.

**Theorem 3.1** (Convergence Rate of Gradient Descent for `ResIST`). *Assume there are $S$ workers, $\ell$ local and $T$ global steps. Assume the depth of the whole ResNet is $H$. Assume for all data indices $i \in [n]$, the data input satisfies $\|\mathbf{x}_i\|_2 = 1$, the data output satisfies $|y_i| = O(1)$, and the number of hidden nodes per layer satisfies $m =$*

$$\Omega\left( \max\left\{ \frac{n^4}{\lambda_{\min}^4\left(\mathbf{K}^{(H)}\right)H^6}, \frac{n^2}{\lambda_{\min}^2\left(\mathbf{K}^{(H)}\right)H^2}, \frac{n}{\delta}, \frac{n^2\log\left(\frac{Hn}{\delta}\right)}{\lambda_{\min}^2\left(\mathbf{K}^{(H)}\right)} \right\} \right).$$

*Set the step size $\eta = O\left( \frac{\lambda_{\min}\left(\mathbf{K}^{(H)}\right)H^2}{n^2\ell^2 S} \right)$ in gradient descent in local training iteration, and follow the procedure as in Algorithm 1. Let the squared-norm loss be*

$L(\theta(t)) := \frac{1}{2}\|\mathbf{y} - f(\theta(t))\|_2^2$, *per $t$ global synchronization round, $t = 1, 2, \ldots T$; here, $\mathbf{y}$ corresponds to the data "labels", and $\theta(t)$ and $f(\theta(t))$ represent the parameters and the output of the whole ResNet, respectively, after $t$-global rounds of* `ResIST`. *Here, $\theta$ includes weights $\mathbf{W}^{(h)}$ at depth $h$ and the last layer's weights $\mathbf{a}$. Then, with probability at least $1 - \delta$ over the random initialization, we have:*

$$L(\theta(t)) \leq \left( 1 - \frac{\eta\ell\lambda_{\min}\left(\mathbf{K}^{(H)}\right)}{2} \right)^t \cdot L(\theta(0)).$$

First, some definitions; more details in the Appendix. Similar to [Du et al., 2019], $\mathbf{K}^{(H)} \in \mathbb{R}^{n \times n}$ is a fixed matrix that depends on the input data, neural network architecture and the activation but does not depend on neural network parameters. Next, we present our method of proving this global result on `ResIST`. Our proof technique is inspired by [Du et al., 2019]: Let the prediction of the network at some $k$-th iteration be $\mathbf{u}(k) = f(\theta(k))$.[3] We formulate the training dynamics as:

$$\mathbf{y} - \mathbf{u}(k+1) = (\mathbf{I} - \eta\mathbf{G}(k))(\mathbf{y} - \mathbf{u}(k)),$$

where $\mathbf{G}_{ij}(k) = \left\langle \frac{\partial u_i(k)}{\partial \theta(k)}, \frac{\partial u_j(k)}{\partial \theta(k)} \right\rangle =$

$$\sum_{h=1}^{H} \left\langle \frac{\partial u_i(k)}{\partial \mathbf{W}^{(h)}(k)}, \frac{\partial u_j(k)}{\partial \mathbf{W}^{(h)}(k)} \right\rangle + \left\langle \frac{\partial u_i(k)}{\partial \mathbf{a}(k)}, \frac{\partial u_j(k)}{\partial \mathbf{a}(k)} \right\rangle$$
$$\triangleq \sum_{h=1}^{H+1} \mathbf{G}_{ij}^{(h)}(k).$$

The proof in [Du et al., 2019] obeys the following ideas: when the width $m$ of deep ResNet is sufficiently large, $\mathbf{G}^{(H)}(k)$ will be very close to $\mathbf{G}^{(H)}(0)$, and all of $\mathbf{G}^{(H)}(k)$'s will be close to the fixed population gram matrix $\mathbf{K}^{(H)}$. The exact definition of $\mathbf{K}^{(H)}$ for ResNet can be found in Section 6 of Du et al. [2019]. Further, $\lambda_{\min}(\mathbf{G}^{(H)}(0))$ is larger than 0. Thus, by standard matrix perturbation analysis, it is shown that $\lambda_{\min}(\mathbf{G}^{(H)}(0))$ is also strictly positive, which will result in linear convergence of deep ResNet.

Here, we further generalize such technique to distributed `ResIST` with layer dropout. The novelty of our proof is that we only conduct gradient descent on sub-ResNets assigned to each local worker. *There is no training iteration with the whole model: this includes the generation of random masks that "champion" parts of the whole ResNet model per worker.* Handling such constructions is the gist of this proof: We carefully analyze the convergence of each subnetwork during local training iterations $\ell$, and prove the global convergence of the combined whole model throughout synchronization rounds $t$. The full proof is provided in section B in the Appendix.

---

[2]Activations of blocks within 3$^{\text{rd}}$ section are still scaled during local SGD pre-training to maintain consistency with `ResIST`.

[3]We use $k$ to abstract the notion of an iteration in [Du et al., 2019]; in our case, a different analysis includes two different iteration indices, $\ell$ and $t$.

# 4 RELATED WORK

Following ResNet, most novel architectures continued to leverage residual connections, which became standard in most architectures. The ResNet architecture has been further modified. *This work focuses on the pre-activation ResNet variant [He et al., 2016a], as it achieves high performance and is well-suited to layer-wise decomposition.*

The focus of this study is on synchronous methods of distributed optimization, such as data parallel training, parallel SGD [Zinkevich et al., 2010], or local SGD [Stich, 2019]. Our methodology is also a variant of model-parallel training [Ben-Nun and Hoefler, 2019, Zhu et al., 2020, Kirby et al., 2020, Gunther et al., 2020, Guan et al., 2019, Chen et al., 2018a]. Many studies have explored possible techniques of synchronous, distributed optimization, yielding a wide number of viable variants [Lin et al., 2018].

To reduce communication costs in the distributed setting, both quantization [Alistarh et al., 2017, Tang et al., 2018] and sparsification [Aji and Heafield, 2017, Jiang and Agrawal, 2018, Wangni et al., 2018, Bouacida et al., 2020] methods have been explored. Similarly, other studies have achieved speedups through the use of low-precision arithmetic during training [Jia et al., 2018]. However, *this line of work is orthogonal to our proposal and can be easily combined with the provided methodology; see section A.5 in the Appendix.*

Large batch training is used to amortize communication and increase throughput for distributed training [Goyal et al., 2017, You et al.]. The properties of large batch training have since been studied extensively [Akiba et al., 2017, You et al., 2017, 2018]. Large batches alter training dynamics, warranting the use of complex heuristics to maintain comparable performance [You et al., 2017]. Here, *we do not focus on the extension of ResIST to the large-batch training domain. Rather, we consider this as future work.*

ResNet robustness to layer removal was explored in [Huang et al., 2016], while [Veit et al., 2016] showed that ensembles of shallow ResNets can yield high performance. [Huang et al., 2016] uses shallow networks during training and scales activations so that all layers may be used for inference. However, our approach is distinct in numerous ways. Primarily, *our method partitions blocks in a stochastic, round-robin fashion, which explicitly prevents the exclusion of layers from training rounds and yields reduced subnetwork depth compared to [Huang et al., 2016].* Inspired by [Li et al., 2017], we also selectively partition residual blocks that are least sensitive to pruning, allowing other layers (i.e., 30% of total layers) to be shared between subnetworks. Unlike [Huang et al., 2016], we avoid partitioning strided layers, which are sensitive to pruning [Li et al., 2017]. Furthermore, our methodology, instead of proposing a form of regularization, focuses on utilizing independent training of shallow sub-ResNets for efficient, distributed training.

Our approach also relates to neural ODE literature. This research connects ResNets as a discrete approximation to a continuous transformation from input to output [Lu et al., 2018]. The neural ODE perspective has been studied both empirically [Chen et al., 2018b, Dupont et al., 2019, Lu et al., 2018] and theoretically [Lu et al., 2020, Thorpe and van Gennip, 2018]. *This provides justification to our approach, as removing ResNet layers can be viewed as approximating the same transformation with a coarser discretization.*

# 5 EXPERIMENTAL DETAILS

Hyperparameters are tuned using a holdout validation set and results are obtained using optimal hyperparameters from the validation set. All experiments are repeated for three trials, and the average performance is presented. We adopt local SGD as our baseline for synchronous, distributed training methods. *In all cases, ResIST achieves comparable performance to local SGD, while lowering the total wall-clock time of training.* We use AWS p3.8xlarge instances for experiments with two or four machines[4] and p3.16xlarge instances for experiments with eight machines. We use each GPU as a single worker that hosts a different sub-ResNet. We assign each worker with a different random seed, so that at each training iteration, it will sample different batches of data.

**Small-Scale Image Classification.** Models are trained with ResIST on CIFAR10 and CIFAR100 for image classification. We adopt standard data augmentation techniques during training and testing [He et al., 2016b]. We adopt a batch size of 128 for each worker. Training is conducted for 80 epochs for experiments with two machines and 160 epochs for experiments with four or eight machines. The recorded performance reflects the best test accuracy achieved throughout training, averaged across three trials. The total wall-clock training time is also reported for each experiment.

**ImageNet Classification.** Models are trained with ResIST on the 1,000-class ILSVRC2012 image classification dataset [Deng et al., 2009]. We adopt standard data augmentation techniques during training and testing, and use a batch size of 256 for each worker [He et al., 2016b]. Training is conducted for 90 epochs. We initialize the learning rate to 0.1 and decrease it $10\times$ at epochs 30 and 60. For all experiments, we set $\ell = 15$, adopt a minimum depth of 10 blocks for each sub-ResNet, and warm-up pre-training using local SGD. For both ResIST and baseline experiments, we utilize momentum restarts and aggregate batch statistics every 1300 synchronization rounds.

**Object Detection.** ResIST is tested in the object detection

---

[4]In section A.4-Appendix and for the Pacal VOC experiment with two machines, we use a cluster with eight V100 GPUs.

domain on the Pascal VOC dataset [Everingham et al., 2010]. Our model, inspired by the Yolo-v2 object detection model [Redmon and Farhadi, 2017], consists of a ResNet101 backbone followed by a detection layer (i.e., a $1 \times 1$ convolution that outputs anchor box predictions). The ResNet backbone of this model is similar to the classification model described in Sec. 2.1, but without the pre-activation structure. The model is trained for 100 epochs with an image dimension of $448 \times 448$ and batch size of 10. No data augmentation techniques are used. The learning rate is increased from $10^{-5}$ to $10^{-4}$ over the first 30 epochs, and decreased by $10\times$ at epochs 60 and 90. Both Pascal VOC 2007 and 2012 training sets are used during training, and performance is evaluated on the Pascal VOC 2007 test set. We report the wall-clock training time and the best loss achieved on the test set throughout training. Experiments are conducted on two and four machines using both local SGD and ResIST.

Table 2: Test accuracy of baseline LocalSGD versus ResIST on small-scale image classification datasets.

|  | # Machines | CIFAR10 | CIFAR100 |
|---|---|---|---|
| Local SGD | 2 | 92.36% ± 0.01 | 70.67% ± 0.03 |
|  | 4 | 92.90% ± 0.06 | 71.51% ± 0.04 |
|  | 8 | 92.00% ± 0.07 | 69.64% ± 0.05 |
| ResIST | 2 | 91.95% ± 0.32 | 70.06% ± 0.51 |
|  | 4 | 92.35% ± 0.22 | 71.30% ± 0.20 |
|  | 8 | 91.45% ± 0.30 | 70.26% ± 0.21 |

# 6 RESULTS

## 6.1 SMALL-SCALE IMAGE CLASSIFICATION

**Accuracy.** The test accuracy on small-scale image classification datasets is listed in Table 2. *ResIST achieves comparable test accuracy in all cases where the same number of machines are used.* ResIST outperforms localSGD on CIFAR100 experiments with eight machines. The performance of ResIST and local SGD are strikingly similar in terms of test accuracy. In fact, the performance gap between the two method does not exceed 1% in any experimental setting. Furthermore, ResIST performance remains stable as the number of sub-ResNets increases, allowing greater acceleration to be achieved without degraded performance (e.g., see CIFAR100 results in Table 2). Generally, using four sub-ResNets yields the best performance with ResIST.

**Efficiency.** In addition to achieving comparable test accuracy, ResIST significantly accelerates training. This acceleration is due to $i$) fewer parameters being communicated between machines and $ii$) locally-trained sub-ResNets being shallower than the global model. Wall-clock training times for four and eight machine experiments are presented in Tables 4. ResIST provides 3.58 to 3.81× speedup in

comparison to local SGD. For eight machine experiments, a significant speedup over four machine experiments is not observed due to the minimum depth requirement and a reduction in the number of local iterations to improve training stability. We conjecture that for cases with higher communication cost at each synchronization and a similar number of synchronizations, eight worker ResIST could lead to more significant speedups in comparison to the four worker case.

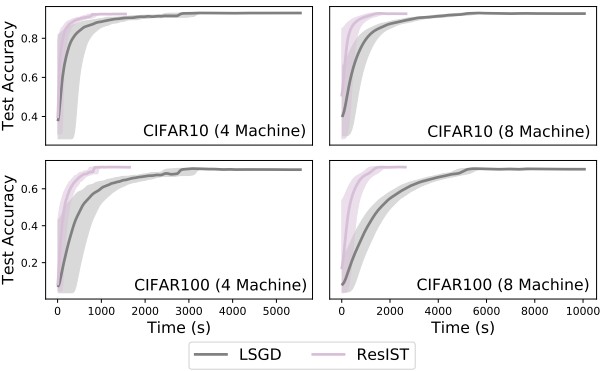

Figure 5: Both methodologies complete 160 epochs of training. Accuracy values are smoothed using a 1-D gaussian filter, and shaded regions represent deviations in accuracy.

A visualization of the speedup provided by ResIST on the CIFAR10 and CIFAR100 datasets is illustrated in Fig. 5. Models trained with ResIST match the final accuracy of those trained with local SGD. Furthermore, increasing the number of sub-ResNets yields an improved speedup for ResIST in comparison to localSGD. It is clear that the communication-efficiency of ResIST allows the benefit of more devices to be better realized in the distributed setting.

## 6.2 LARGE-SCALE IMAGE CLASSIFICATION

**Accuracy.** The test accuracy of models trained with both ResIST and local SGD for different numbers of machines on the ImageNet dataset is listed in Table 3. As can be seen, *ResIST achieves comparable test accuracy ($< 2\%$ difference) to local SGD in all cases.* Additionally, as shown in [Recht et al., 2019], many current image classification models overfit to the ImageNet test set and cannot generalize well to new data. Thus, models trained with both local SGD and ResIST are also evaluated on three different Imagenet V2 testing sets [Recht et al., 2019]. As shown in Table 3, ResIST consistently achieves comparable test accuracy in comparison to local SGD on these supplemental test sets.

**Efficiency.** As shown in Tables 3 and 5, ResIST significantly accelerates the ImageNet training process. However, due to the use of fewer local iterations and the local SGD warm-up phase, the speedup provided by ResIST is smaller relative to experiments on small-scale datasets. In Table 3, ResIST can reduce the total communication volume during

Table 3: Performance of baseline models and models trained with `ResIST` on 1K Imagenet [Recht et al., 2019]. MF stands for test set "MatchedFrequency" and was sampled to match the MTurk selection frequency distribution of the original ImageNet validation set; T-0.7 stands for test set "Threshold0.7" and was built by sampling ten images for each class among the candidates with selection frequency at least 0.7; TI stands for test set "TopImages" and contains the ten images with highest selection frequency for each class.

| | # Machines | Imagenet | Imagenet V2 Test Set | | | Training Time | Speedup | Communication | Cost Ratio |
| | | | MF | T-0.7 | TI | | | | |
|---|---|---|---|---|---|---|---|---|---|
| Local SGD | 2 | 73.32% | 60.72% | 69.47% | 75.48% | 48.61 hours | - | 7546.80 GB | - |
| | 4 | 72.66% | 59.88% | 68.34% | 74.27% | 29.29 hours | - | 7546.80 GB | - |
| `ResIST` | 2 | 71.60% | 58.92% | 67.51% | 73.56% | 36.79 hours | **1.32**× | 5831.2 GB | **1.29**× |
| | 4 | 70.74% | 57.56% | 66.46% | 72.65% | 22.37 hours | **1.31**× | 6007.6 GB | **1.26**× |

Table 4: Training time in seconds of baseline models and models trained with `ResIST` on small-scale image classification datasets.

| | # Machines | Dataset | Total Time | Speedup |
|---|---|---|---|---|
| Local SGD | 4 | C10 | 5486 ± 7.05 | - |
| | | C100 | 5528 ± 65.90 | - |
| | 8 | C10 | 10072 ± 5.12 | - |
| | | C100 | 10058 ± 8.71 | - |
| `ResIST` | 4 | C10 | 1532 ± 0.83 | **3.60**× |
| | | C100 | 1545 ± 1.27 | **3.58**× |
| | 8 | C10 | 2671 ± 3.25 | **3.77**× |
| | | C100 | 2639 ± 3.89 | **3.81**× |

training, which is an important feature in the implementation of distributed systems with high computational costs.

Table 5: Total training time on Imagenet (in hours) of models trained with both local SGD and `ResIST` using two and four machines to reach a fixed test accuracy.

| # Machines | Target Accuracy | Local SGD | `ResIST` | Speedup |
|---|---|---|---|---|
| 2 | 71.00 | 33.26 | 26.63 | **1.25**× |
| 4 | 70.70 | 18.50 | 18.12 | **1.02**× |

## 6.3 OBJECT DETECTION

**Loss.** The test loss of models trained with both `ResIST` and local SGD for different numbers of machines on the Pascal VOC object detection dataset is listed in Table 6. Notably, `ResIST` achieves a lower test loss in comparison to local SGD for the experiment with two machines. Although the test loss achieved by `ResIST` is slightly worse than local SGD in the four machine case, the performance is comparable. Namely, the difference in test loss achieved by local SGD and `ResIST` never exceeds a value of one.

**Efficiency.** In addition to achieving comparable or improved test loss in comparison to local SGD, `ResIST` also provides

a significant training acceleration on the PascalVOC dataset. In particular, models trained with `ResIST` achieve up to a $1.64\times$ acceleration in comparison to object detection models trained with localSGD.

Table 6: Test loss and total training time in seconds on Pascal VOC for models trained with both local SGD and `ResIST` using two and four machines. Training time in seconds.

| | # Machines | Test Loss | Train Time | Speedup |
|---|---|---|---|---|
| Local SGD | 2 | 6.15 ± 0.03 | 39621 ± 9.12 | - |
| | 4 | 6.22 ± 0.06 | 16840 ± 0.11 | - |
| `ResIST` | 2 | 5.99 ± 0.01 | 24058 ± 3.22 | **1.64**× |
| | 4 | 6.69 ± 0.17 | 11264 ± 49.38 | **1.49**× |

## 6.4 MORE EXPERIMENTS

In the Appendix A, we outline numerous ablation experiments that were performed using `ResIST`. These experiments provide an understanding of the algorithm's behavior, as well as empirical support for its design: they include `ResIST` design decisions (section A.1), comparison of `ResIST` with ensemble methods (section A.2), robustness to local iterations (section A.3), applicability of `ResIST` to deeper architectures (section A.4), and compatibility to existing quantization/sparsification techniques (section A.5).

## 7 CONCLUSION

In the work, we present `ResIST`, a novel algorithm for synchronous, distributed training of ResNets. `ResIST` operates by decomposing a global ResNet model into several shallower sub-ResNets, which are trained independently and itermittently aggregated into the global model. By only communicating parameters of sub-ResNets between machines and training shallower, less expensive networks, `ResIST` reduces the communication and local training cost of synchronous, distributed training. We demonstrate the impact of `ResIST` on several image classification datasets, as well

as in the object detection domain, by highlighting the significant training acceleration it provides in comparison to methods like local SGD [Lin et al., 2018] without any deterioration in performance.

We aim to extend `ResIST` to other network architectures, as `ResIST` is fully-extensible to all network architectures with residual connections. Because residual connections are now standard in most important deep learning architectures (e.g., transformers), many opportunities to extend applications of `ResIST` exist. On the other hand, `ResIST` has been shown to be fully-compatible with various gradient compression methods. As such, we will investigate the prospect of fully integrating such compression methods within `ResIST`, both during training and communication phases, to further decrease memory and computation costs.

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
