# OpenReview forum: "ResIST: Layer-Wise Decomposition of ResNets for Distributed Training"
_auai.org/UAI/2022/Conference — UAI 2022 Poster_

### Official Review · Reviewer_1zbs · 2022-03-23

**Q2(1) Originality/Novelty:** 1
**Q2(2) Significance/Impact:** 2
**Q2(3) Correctness/Technical Quality:** 3
**Q2(6) Clarity Of Writing:** 3
**Q6 Overall Score:** 3
**Q8 Confidence In Your Score:** 3

**Q1 Summary And Contributions:**

This paper proposes ResIST, a distributed training protocol that decomposes the global ResNet into small sub-models and trains them locally and independently, in order to reduce the communication overhead and memory footprint. Theoretical analysis proves the convergence of ResIST. Experiments show good performance compared to local SGD.

**Q2 Assessment Of The Paper:**

More detailed information regarding each of these aspects is given below:

**Q2(4) Quality Of Experiments (Optional):**

3: Good: The experimental evaluation is adequate, and the results convincingly support the main claims.

**Q2(5) Reproducibility:**

3: Good: Key resources (e.g., proofs, code, data) are available and key details (e.g., proofs, experimental setup) are sufficiently well-described for competent researchers to confidently reproduce the main results.

**Q3 Main Strengths:**

1. The overall idea is technically sound. ResIST reduces not only the communication overhead, but also the memory footprint for each worker.
2. Theoretical analysis proves the convergence of ResIST.
3. Experiments show good performance compared to local SGD.

**Q4 Main Weakness:**

1. My main concern is that the overall idea is possibly not novel. The general idea (partition model/layer into sub-model/blocks and train locally) sounds very similar to the following paper:
Bouacida, Nader et al. “Adaptive Federated Dropout: Improving Communication Efficiency and Generalization for Federated Learning.” IEEE INFOCOM 2021 - IEEE Conference on Computer Communications Workshops (INFOCOM WKSHPS) (2021): 1-6.
If that is true, than the novelty and contribution of this paper may not be enough for publication.
2. This paper is limited to ResNet.

**Q5 Detailed Comments To The Authors:**

1. (Minor) I found that the baseline without any communication compression could be found in Appendix B (I suppose "vanilla data parallel parallel" is the same as "vanilla SGD"). I would recommend to put this baseline in the main paper. However, as long as this baseline is included somewhere, I don't think it's a real issue. BTW, there is a typo in the legend of Figure 8: Vanila -> Vanilla
2. My main concern is that the overall idea is possibly not novel. The general idea (partition model/layer into sub-model/blocks and train locally) sounds very similar to the following paper:
Bouacida, Nader et al. “Adaptive Federated Dropout: Improving Communication Efficiency and Generalization for Federated Learning.” IEEE INFOCOM 2021 - IEEE Conference on Computer Communications Workshops (INFOCOM WKSHPS) (2021): 1-6.
Federated Dropout also partition the layers into blocks and train them locally. Although Federated Dropout is based on federated learning, it is well-known that federated learning is very similar to local SGD. Furthermore, ResIST uses random partitioning, while Federated Dropout goes beyond random: it's using sth. called "activation score" to select the sub-models intentionally.
Please provide a detail comparison between this paper and Federated Dropout. If the authors can show that the idea and algorithm of this paper is significantly different from Federated Dropout, I would be happy to raise the score.


**Q7 Justification For Your Score:**

The main reason for reject is that the overall idea of this paper is very similar to Federated Dropout, which significantly weakens the novelty and contribution.

**Q9 Complying With Reviewing Instructions:**

1: Yes.

---

### Official Review · Reviewer_V3V1 · 2022-03-28

**Q2(1) Originality/Novelty:** 2
**Q2(2) Significance/Impact:** 2
**Q2(3) Correctness/Technical Quality:** 3
**Q2(6) Clarity Of Writing:** 2
**Q6 Overall Score:** 6
**Q8 Confidence In Your Score:** 3

**Q1 Summary And Contributions:**

The paper proposes ResIST, a technique for decomposing ResNets, that allows for more efficient distributed training by reducing the communication requirements, compared to other distributed training methods; it also accelerates local training since the deep global model is decomposed into shallower ones.
The method is tested on CIFAR 10,100, on Imagenet and on object detection. Theory is provided establishing convergence over the global synchronisation moves.

**Q2 Assessment Of The Paper:**

More detailed information regarding each of these aspects is given below:

**Q2(4) Quality Of Experiments (Optional):**

3: Good: The experimental evaluation is adequate, and the results convincingly support the main claims.

**Q2(5) Reproducibility:**

2: Fair: Key resources (e.g., proofs, code, data) are unavailable but key details (e.g., proof sketches, experimental setup) are sufficiently well-described for an expert to confidently reproduce the main results.

**Q3 Main Strengths:**

The method seems to be performing well empirically with test accuracy close to the local SGD, the benchmark, with training time and communication reduced by factors up to 4 in some cases. The method is compared to ensembles of shallow networks, which completely bypass communication but need to store more parameters. Although I have not read the proof in full detail, the parts I read seem mostly correct.


**Q4 Main Weakness:**

Some aspects of the writing can be improved. The method is never described in full detail; Algorithm 1 is not really informative. It's not clear for example whether each subnetwork is trained on the full data set or only on a subset. A more thorough, theoretical discussion of the communication cost gains seems to be missing; I guess this could be added to the supplementary without too much work.
The convergence rate in Theorem 3.1 is over the number of global synchronisation steps. Given that global synchronisation moves happen relatively rarely to keep costs down, this would slow down the convergence. Also, there are some smoothness assumptions on the activation function, which I guess have appeared fairly often in the literature; they are not ideal but we can live with them.

**Q5 Detailed Comments To The Authors:**

1. Please read carefully the whole manuscript to correct typos, grammar and syntax. e.g. Lemma B.5 the assumptions on $\sigma$ are repeated twice *in Section B.1 syncrhonization, etc.
2. The training dynamics in section 3 are formulated as $y-u(k+1)=(I-\eta G(k)) (y-u(k))$. To my understanding this is just an ansatz derived from the continuous gradient flow that provided some intuition. If this is the case it could be explained.
3. In Section B.1 there are symbols $\mathbf{u}(t), \hat{\mathbf{u}}(t), \hat{u}(t)$. These all seem to refer to the output of the NN at global sync step $t$. The arguments $\hat{\mathbf{u}}$ takes also vary, look for example at Condition B.1
4. Condition B.1: as far as I understand this is a claim that is used in the proof and justified later. Being stated as a condition is a little confusing.
5.I find Algorithm 1 to not be very informative. Maybe you could explain the full algorithm in the supplementary in more detail; for example each subnetwork is being trained on the full data or not? For the proof sketch it seems so. A c

**Q7 Justification For Your Score:**

I am leaning towards an accept. The paper is good, without any major flaws, presents a new method which does offer computational advantages. The presentation needs improvement and could make a significant difference. The gains in terms of communication and training costs, are not huge but they could make a difference in some cases. Therefore there is potential for impact.

**Q9 Complying With Reviewing Instructions:**

1: Yes.

---

### Official Review · Reviewer_JWxD · 2022-04-13

**Q2(1) Originality/Novelty:** 2
**Q2(2) Significance/Impact:** 3
**Q2(3) Correctness/Technical Quality:** 3
**Q2(6) Clarity Of Writing:** 3
**Q6 Overall Score:** 6
**Q8 Confidence In Your Score:** 4

**Q1 Summary And Contributions:**

In this paper, the authors proposed ResIST, which randomly decomposes a global ResNet into several sub-ResNets that are trained independently, before having their updates synchronized and aggregated into the global model. In the next round, new sub-ResNets are randomly generated and the process repeats until convergence. ResIST reduces the per-iteration communication, memory, and time requirements of ResNet training to only a fraction of the requirements of full-model training.


**Q2 Assessment Of The Paper:**

More detailed information regarding each of these aspects is given below:

**Q2(4) Quality Of Experiments (Optional):**

3: Good: The experimental evaluation is adequate, and the results convincingly support the main claims.

**Q2(5) Reproducibility:**

3: Good: Key resources (e.g., proofs, code, data) are available and key details (e.g., proofs, experimental setup) are sufficiently well-described for competent researchers to confidently reproduce the main results.

**Q3 Main Strengths:**

proposed a new layer-wise model decomposition for distributed training and achieve good empirical performances

**Q4 Main Weakness:**

 theoretical result is not clearly presented and it seems that a bunch of tricks are the key for performance improvements.



**Q5 Detailed Comments To The Authors:**

1 . Why do the authors restrict this framework on dealing with ResNet only? It seems to me that the proposed method can also work on other types of neural net architectures. Is there any particular reason for doing so?

2. Maybe a miss this somewhere but it seems that the proposed method would require each worker to have the entire training dataset? (since most layers will only be trained by one worker). That seems to be another potential downside? What if the data is also distributed in different workers? Can the proposed method still work? What if the data is non-iid distributed?

3 . The theoretical result is not clearly presented. Specifically, there is no introduction to what are the assumptions (smoothness, maybe?) used for the analysis and no discussion about whether these settings are practical or aligned with the real world scenarios. And it is hard to draw clear conclusions from the theoretical result.

4. The following recent work also discussed communication compression strategies in the distributed setting. The author might also want to comment on them.

"EF21: A new, simpler, theoretically better, and practically faster error feedback." NeurIPS2021.
"Communication-Compressed Adaptive Gradient Method for Distributed Nonconvex Optimization." AISTATS2022

**Q7 Justification For Your Score:**

good empirical performances but the theoretical part is not clearly written

**Q9 Complying With Reviewing Instructions:**

1: Yes.

---

### Decision · Program_Chairs · 2022-05-15

**Decision:**

Accept (Poster)

**Comment:**

Meta Review: The paper develops a distributed training algorithm for residual networks. The approach works by randomly breaking down a resnet into sub-resnets that are trained separately before having their updates synchronized. The approach reduces communication overhead and memory footprint. The reviewers were generally positive (as am I) except for one concern about the potential novelty of the work raised by reviewer 1zbs. This novelty question was thoroughly addressed in the author response to this reviewer. When asked how the positive reviewers felt in light of this concern both reviewers wanted to keep their opinion as is. I am supportive of this paper with the noted limitation that the approach is only for the ResNet.